# Chloroplast Microsatellite-Based High-Resolution Melting Analysis for Authentication and Discrimination of *Ilex* Species

Yonguk Kim [ID], Dool-Ri Oh [ID], Yu-Jin Kim, Kyo-Nyeo Oh and Donghyuk Bae *

Jeonnam Institute of Natural Resources Research, Jangheung-gun 59338, Jeollanam-do, Korea
* Correspondence: bdhyuch@naver.com

**Abstract:** *Ilex* species are important sources of high-quality raw plant materials for the production of drugs and functional foods. The precise identification of different species within the *Ilex* genus would greatly facilitate authentication and certification as well as forest resource monitoring in plantations. Combining DNA barcoding with high-resolution melting (HRM) analysis represents a robust strategy for species discrimination, as demonstrated in recent DNA barcoding studies. Here, using concatenated and aligned complete chloroplast genomes of different *Ilex* species, we conducted a sliding window analysis to identify regions of high nucleotide diversity (*Pi*). We optimized and validated the utility of PCR-based HRM coupled with microsatellite markers to discriminate among the four *Ilex* species, *Ilex integra* Thunb., *Ilex rotunda* Thunb., *Ilex cornuta* Lindl. and Paxton, and *Ilex x wandoensis* C.F. Mill and M. Kim, from wild populations in southwestern Korea. The marker *trnS^{UGA}*-*psbZ* produced clear melting patterns and distinct melting curve profiles for the four *Ilex* species using HRM analysis. We applied this protocol to commercially available *Ilex* accessions and consistently identified the correct species for all 15 accessions tested. Therefore, combining DNA barcoding with HRM analysis is a powerful method for identifying different species within the same genus, which could be used for quality control of raw materials in the functional food/medicinal plant industry.

**Keywords:** *Ilex* species; *Ilex integra*; *Ilex rotunda*; *Ilex cornuta*; *Ilex x wandoensis*; HRM; microsatellite marker; DNA barcoding

## 1. Introduction

*Ilex* species, a monotypic genus in the family Aquifoliaceae, are dioecious trees or shrubs with either evergreen or deciduous habits that are found in tropical and subtropical regions such as South America and Southeast Asia. There are more than 600 *Ilex* species, including *I. paraguariensis* A. St.-Hil., *I. cornuta*, and *I. rotunda*, which are valuable materials for the health food, cosmetics, and pharmaceutical industries [1–4]. Yerba mate, an infusion prepared from the leaves of *I. paraguariensis*, is widely consumed worldwide. Mate is produced in tropical countries such as Paraguay, Argentina, and Brazil, and has long been used as South American folk medicine. Many studies have confirmed the therapeutic efficacy of a mate by evaluating its bioactivity *in vitro* and *in vivo*. As a result, mate has emerged as a major source of valuable bioactive components that are effective in preventing arthritis, inflammatory diseases, cardiovascular diseases, diabetes, and obesity [5–9]. Therefore, *I. paraguariensis* has been recognized as a popular product by the health and wellness industries worldwide. These findings have prompted research on other species of the genus *Ilex*. Recent studies have demonstrated the antioxidant and cardioprotective effects of triterpenoid saponins from *I. kaushue* S.Y. Hu and *I. cornuta* [3,10]. *I. rotunda* extracts showed enhanced anti-inflammation activity in vitro and prevented obesity in vivo [11,12].

In Korea, in an effort to respond to changes in woody vegetation due to climate change, many ecologists and botanists are focusing on the conservation and development of warm temperate evergreen broad-leaved trees such as *Ilex* (Aquifoliaceae), *Quercus*

(Fagaceae), *Castanopsis* (Fagaceae), *Camellia* (Theaceae), *Cinnamomum* (Lauraceae), and *Machilus* (Lauraceae) [13,14]. Approximately six species of the genus *Ilex* are native to Korea, including *I. integra*, *I. rotunda*, and *I. cornuta*, *I. crenata* Thunb, *I. crenata* var. *microphylla* Maxim. ex Matsum. *I. latifolia* Thunb., *I. x wandoensis*, and *I. macropoda* Miq. Four of these species (*I. integra*, *I. rotunda*, *I. cornuta*, and *I. x wandoensis*) are found on islands and in coastal areas in the south. *I. x wandoensis*, which was discovered on Wando Island in Korea, originated from the natural hybridization of the parent species *I. cornuta* and *I. integra* [15]. This hybrid species shares several characteristics with its two parents, including evergreen leaves, dioecy, and red drupes [16]. The leaf shape of *I. x wandoensis* is intermediate between that of *I. cornuta* (with sharp thorns on the leaf margins) and *I. integra* (with an ovate or elliptical ovate shape). It is difficult to discriminate *I. x wandoensis* from *I cornuta* based on their morphological characteristics alone. Moreover, *I. cornuta* and *I. x wandoensis* both have very sharp and spiny leaf margins, whereas *I. integra* and *I. rotunda* have entire leaf margins (Figure 1).

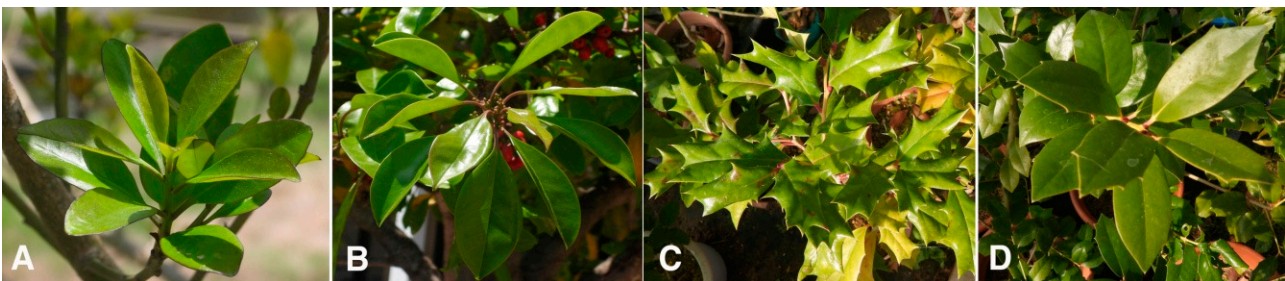

**Figure 1.** Typical leaf shape of *I. integra* (**A**), *I. rotunda* (**B**), *I. cornuta* (**C**), and *I. x wandoensis* (**D**).

Plant DNA barcoding using markers based on ribosomal internal transcribed spacer (ITS) and chloroplast universal regions, including *matK*, *rbcL*, and *trnH-psbA*, is a common method for species identification and classification. However, it is difficult to accurately distinguish different species from a single plant genus using a few core DNA barcode regions; discriminating between closely related species or sub-species is particularly challenging [17,18]. Therefore, a complementary method is needed using a specialized DNA barcode region to distinguish closely related species or similar species from the same genus. Complete chloroplast genome sequences can now be easily and accurately assembled at a relatively low cost due to rapid advances in the development of sequencing platforms. In addition, more than 5000 complete chloroplast genome sequences deposited in the National Center for Biotechnology Information (NCBI) organelle genome database can be used to develop microsatellite markers by comparative sequence analysis. Many studies have demonstrated the accurate discrimination of closely related species within the same genus using species-specific barcodes derived from a comparative analysis of complete chloroplast genomes [19–21]. The use of these super DNA barcodes based on whole chloroplast genome sequences offers a solution to the limitations of using previously established core DNA barcodes based on short gene or intergenic spacer (IGS) regions (<1 kb) to clearly discriminate among closely related species within the same genus.

High-resolution melting (HRM) analysis is a method used to visualize significantly different melting temperatures (Tm) and peak locations between genotypes or alleles based on quantitative PCR [21]. DNA barcoding coupled with HRM (Bar-HRM) analysis was recently shown to be an efficient molecular tool for the correct species identification and authentication of herbal medicinal plants and for the accurate quantification of adulterants in commercial natural food products [21–23].

In the current study, we searched for differences in the complete chloroplast genome sequences of various *Ilex* species in order to identify a new DNA barcode for *Ilex* species. Our goal was to develop a chloroplast genome-based Bar-HRM marker for the rapid and accurate authentication of four *Ilex* species with similar morphological features.

## 2. Materials and Methods

### 2.1. Plant Materials and DNA Extraction

The plant materials used in this study were leaf tissue from five accessions of *I. integra*, *I. rotunda*, and *I. cornuta*, which were obtained from the National Institute of Biological Resources (NIBR), Republic of Korea, and leaf tissue from 15 accessions collected from Wando Arboretum (Wando Island, Republic of Korea) (Table 1). To extract total genomic DNA, fresh leaf samples from all accessions (80 mg wet weight) were placed in microfuge tubes filled with stainless steel beads (2.8 mm in diameter) from a DNeasy Plant Pro Kit (Qiagen, Valencia, CA, USA). The leaves were ground to powder in a Precellys® Evolution homogenizer (Bertin Technologies, Montigny-le-Bretonneux, France). DNA extraction was performed using a DNeasy Plant Pro Kit according to the manufacturer's instructions. DNA concentration and quality were measured using a NanoDrop 2000c spectrophotometer (Thermo Fisher Scientific, Waltham, MA, USA), and DNA integrity was evaluated by agarose gel electrophoresis on a 1.5% (*w/v*) agarose gel. An aliquot of DNA was diluted in nuclease-free water to a final working stock of 10 ng/μL.

**Table 1.** *Ilex* specimens used in this study.

| No. | Scientific Name | Common Name | Collection Site | Specimen Code |
|---|---|---|---|---|
| 1 | | | 34°21′37.8″ N126°39′51.2″ E | NIBRGR0000630040 |
| 2 | | | 34°21′37.5″ N 126°39′51.5″ E | JINR000003111 |
| 3 | *Ilex integra* Thunb. | *Kamtangnamu* | 34°21′37.4″ N 126°39′52.2″ E | JINR000003112 |
| 4 | | | 34°21′37.3″ N 126°39′52.5″ E | JINR000003113 |
| 5 | | | 34°41′12.1″ N 126°53′47.6″ E | JINR000003114 |
| 6 | | | 33°18′09.4″ N 126°18′47.9″ E | NIBRGR0000429868 |
| 7 | | | 34°41′11.8″ N 126°53′48.8″ E | JINR000003115 |
| 8 | *Ilex rotunda* Thunb. | *Meonnamu* | 34°21′44.5″ N 126°39′43.9″ E | JINR000003116 |
| 9 | | | 34°21′44.4″ N 126°39′43.7″ E | JINR000003117 |
| 10 | | | 34°21′44.2″ N 126°39′44.0″ E | JINR000003118 |
| 11 | | | 33°20′12.7″ N 126°12′03.5″ E | NIBRGR0000429502 |
| 12 | | | 33°20′00.6″ N 126°10′40.6″ E | NIBRGR0000630049 |
| 13 | *Ilex cornuta* Lindl.andPaxton | *Horanggasinamu* | 33°18′55.3″ N 126°10′33.4″ E | NIBRGR0000630051 |
| 14 | | | 35°00′23.2″ N 126°49′21.5″ E | JINR000003119 |
| 15 | | | 35°00′23.3″ N 126°49′24.3″ E | JINR000003120 |
| 16 | | | 34°55′38.1″ N 127°30′06.6″ E | JINR000003121 |
| 17 | | | 35°00′23.5″ N 126°49′20.1″ E | JINR000003122 |
| 18 | *Ilex x wandoensis* C.F. Mill. and M. Kim | *Wandohoranggasinamu* | 35°00′23.5″ N 126°49′20.2″ E | JINR000003123 |
| 19 | | | 34°21′38.3″ N 126°39′51.4″ E | JINR000003124 |
| 20 | | | 34°21′38.1″ N 126°39′51.8″ E | JINR000003125 |

### 2.2. Comparison of Chloroplast Genomes and Identification of Microsatellite Loci

All six available complete chloroplast genome sequences of the four *Ilex* species were downloaded from the NCBI GenBank database (*I. integra*, NC_044417 and MK335537; *I rotunda*, MT764244; *I. cornuta*, NC_044016 and MK335536; and *I. x wandoensis*, OK328174). The chloroplast genome sequences were aligned using the ClustalW algorithm from MEGA 7.0 [24]. The mVISTA program (http://genome.lbl.gov/vista/mvista/submit.shtml, accessed on 25 August 2021) was used in Shuffle-LAGAN mode to compare the six *Ilex* chloroplast genomes, using the *I. integra* (NC_044417) chloroplast genome as a reference. CP-GAVA2 was used to annotate the chloroplast genomes and to predict junctions among these four *Ilex* species, which were visualized using IRscope (http://irscope.shinapps.io./irapp/, accessed on 21 March 2022) based on the annotations of their chloroplast genomes in GenBank. The six *Ilex* chloroplast genomes were aligned with MAFFT version 7.480 [25] and scanned for regions of high nucleotide diversity ($\pi$) by sliding window analysis implemented in DNaSP version 6.0 [26] with a window and step size of 600 bp. Only regions with a nucleotide diversity ($\pi$) value of >0.08 were considered to be hotspot regions. Each

hotspot region was visually identified as a single nucleotide polymorphism (SNP), simple sequence repeat (SSR), or insertion/deletion (InDel) using microsatellite markers to discriminate among the four species. The NCBI BLAST analysis tool was employed to identify and confirm phylogenetically species-specific regions for each candidate marker. The goal was to discover species-specific sequences of the four *Ilex* species. The locus sequences of the four *Ilex* chloroplast genomes were collected by multiple sequence alignment, and each locus sequence isolated from the *I. integra* chloroplast genome was further evaluated via an NCBI BLASTn search to confirm the similarities and differences among sequences of *I. integra, I. rotunda, I cornuta*, and *I. x wandoensis*.

### 2.3. Quantitative PCR and HRM Analysis of Candidate DNA Barcodes

To validate polymorphisms derived from the complete cp genomes of the four *Ilex* species, species-specific primers were designed using Primer Tool software from Integrated DNA Technologies (IDT, Coralville, IA, USA) based on the hotspot regions identified above, except for multi-copy genes identified in these *Ilex* chloroplast genomes. This software uses standard design parameters for HRM, such as primer length (23–25 nucleotides), melting temperature (58–60 °C), GC content (40%–50%), and amplicon size (<200 bp) to design optimal primer pairs. To ensure the specificity of our primers, a Primer-BLAST analysis was conducted on the NCBI site to compare each primer against the genome sequences of the *Ilex* genus. All oligonucleotides used for quantitative PCR were synthesized at Cosmo Genetech (Seoul, Korea). A BioRad CFX96 instrument (Bio-Rad Laboratories, Hercules, CA, USA) was used for quantitative PCR and the HRM assay. Each 20-μL reaction contained 1 μL of genomic DNA (10 ng), 5 pM of primers, and 10 μL of reaction buffer, which included IQ SyberGreen Supermix (Bio-Rad). The PCR conditions were as follows: initial denaturation at 95 °C for 2 min; 40 cycles of denaturation (95 °C for 10 s), annealing (60 °C for 30 s), and extension (72 °C for 30 s), with the fluorescence signal collected at the end of the annealing and extension steps. Data were processed using Bio-Rad CFX Manager 3.1 software (Bio-Rad Laboratories, Hercules, CA, USA). PCR amplicons were denatured for HRM at 95 °C for 30 s and then annealed at 60 °C for 1 min to form random DNA duplexes. These steps were followed by a melting curve from 65 °C to 95 °C with an increase of 0.2 °C every 10 s. Fluorescence data were obtained at the end of each melting phase and processed using Precision Melt Analysis Software 1.2 (Bio-Rad Laboratories, Hercules, CA, USA), producing melting curves as a function of temperature and difference curves for easy visual identification of clusters.

### 3. Results

### 3.1. Comparative Analysis of the Chloroplast Genome of Four Ilex Species

We compared the features of the complete chloroplast genomes of *I. integra* (NC_044417 and MK335537), *I. rotunda* (MT764244), *I. cornuta* (NC_044416 and MK335536), and *I. x wandoensis* (OK328174), which are readily available in NCBI GenBank. The lengths of these cp genomes ranged from 157,216 bp (*I. x wandoensis*) to 157, 780 bp (*I. rotunda*). The lengths of the large single-copy (LSC) region ranged from 86,607 bp (*I. x wandoensis*) to 87,094 bp (*I. rotunda*), while the small single-copy (SSC) regions ranged from 18,426 bp (*I. integra*) to 18,436 bp (*I. rotunda*). The pairs of inverted repeat (IR) regions ranged from 52,182 bp (*I. cornuta* and *I. x wandoensis*) to 52,250 bp (*I. rotunda*). All six chloroplast genomes of the four *Ilex* species contained a total of 131 genes, consisting of 86 protein-coding genes, 37 transfer RNA (tRNA) genes, and eight ribosomal RNA (rRNA) genes. The overall GC content in each cp genome varied little, from 37.6 (*I. integra* and *I. rotunda*) to 37.7% (*I. cornuta* and *I. x wandoensis*) (Table 2). The chloroplast genomes of the four *Ilex* species had the same gene order and gene arrangement.

**Table 2.** Genes in the chloroplast genomes of the four *Ilex* species. [a] Genes with two copies; * Genes with one intron; ** Genes with two introns.

| Category | Gene Group | Gene Name | | | | |
|---|---|---|---|---|---|---|
| Self-replication | Ribosomal RNA genes | *rrn4.5* [a] | *rrn5* [a] | *rrn16* [a] | *rrn23* [a] | |
| | Transfer RNA genes | *trnA-UGC* * | *trnC-GCA* | *trnD-GUC* | *trnE-UUC* | *trnF-GAA* |
| | | *trnfM-CAU* | *trnG-GCC* | *trnG-UCC* * | *trnH-GUG* | *trnI-CAU* [a] |
| | | *trnI-GAU* [a],* | *trnK-UUU* * | *trnL-CAA* [a] | *trnL-UAA* * | *trnL-UAG* |
| | | *trnM-CAU* | *trnN-GUU* [a] | *trnP-UGG* | *trnQ-UUG* | *trnR-ACG* [a] |
| | | *trnR-UCU* | *trnS-GCU* | *trnS-UGA* | *trnS-GGA* | *trnT-UGU* |
| | | *trnT-GGU* | *trnV-GAC* [a] | *trnV-UAC* * | *trnV-GAC* | *trnW-CCA* |
| | | *trendy-GUA* | | | | |
| | Small ribosome subunit | *rps2* | *rps3* | *rps4* | *rps7* [a] | *rps8* |
| | | *rps11* | *rps12A* * | *rps12B* [a],** | *rps14* | *rps15* |
| | | *rps16* * | *rps18* | *rps19* | | |
| | Large ribosome subunit | *rpl2* [a],* | *rpl14* | *rpl16* * | *rpl20* | *rpl22* |
| | | *rpl23* [a] | *rpl32* | *rpl33* | *rpl36* | |
| | RNA polymerase | *rpoA* | *rpoB* | *rpoC1* * | *rpoC2* | |
| | Translation initiation factor | *infA* | | | | |
| Photosynthesis | Photosystem I subunit | *psaA* | *psaB* | *psaC* | *psaI* | *psaJ* |
| | | *ycf3* ** | *ycf4* | | | |
| | Photosystem II subunit | *psbA* | *psbB* | *psbC* | *psbD* | *psbE* |
| | | *psbF* | *psbH* | *psbI* | *psbJ* | *psbK* |
| | | *psbL* | *psbM* | *psbT* | *psbZ* | |
| | Cytochrome *b/f* complex | *petA* | *petB** | *petD* * | *petG* | *petL* |
| | | *petN* | | | | |
| | ATP synthase | *atpA* | *atpB* | *atpE* | *atpF** | *atpH* |
| | | *atpI* | | | | |
| | Rubisco large chain | *rbcL* | | | | |
| | NADH dehydrogenase | *ndhA* * | *ndhB* [a],* | *ndhC* | *ndhD* | *ndhE* |
| | | *ndhF* | *ndhG* | *ndhH* | *ndhI* | *ndhJ* |
| | | *ndhK* | | | | |
| Other genes | Maturase | *matK* | | | | |
| | Membrane protein | *cemA* | | | | |
| | Acetyl-CoA carboxylase | *accD* | | | | |
| | Cytochrome *c* biogenesis | *ccsA* | | | | |
| | ATP-dependent protease | *clpP* ** | | | | |
| | TIC complex component | *ycf1* [a] | | | | |
| Genes of unknown functions | Open reading frames | *ycf2* | *ycf15* | | | |

We compared the border structures of the four chloroplast genomes in detail (Figure 2). IR regions contained *rpl22*, and the SSC/IRa and IRb/SSC borders contained parts of the *ycf1* and *ndhF* genes, respectively. The *I. rotunda* chloroplast genome harbored two copies of the *ycf1* gene, including one in the IRb/SSC border. The *ndhF* gene was located in the IRb/SSC boundary region at a distance of 15 to 2244 bp from the borders.

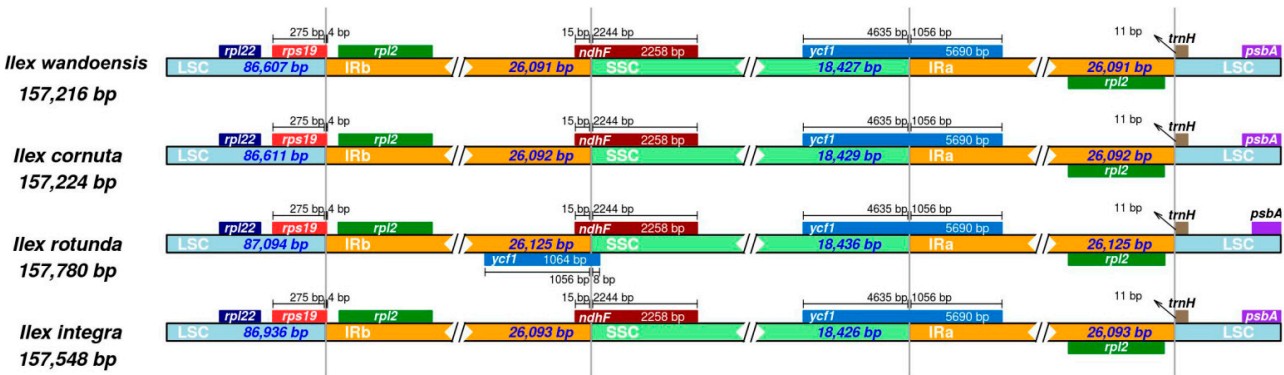

**Figure 2.** Distances between adjacent genes and junctions of the large single-copy (LSC), small single-copy (SSC), and two inverted-repeat (IR) regions among the chloroplast genomes of the four *Ilex* species.

### 3.2. Divergence Hotspots in the Four Ilex Cp Genomes

We compared the sequence divergence among the four *Ilex* chloroplast genomes using mVISTA, using the *I. integra* annotation as the reference (Figure 3). The IR (A/B) regions were less divergent than the LSC and SSC regions. In addition, the non-coding regions were more variable than the coding regions, and the most highly divergent regions among the four chloroplast genomes occurred in the intergenic spacers (IGS). To determine the level of sequence divergence, we calculated the nucleotide variability ($\pi$) values for regions spanning 300 bp on either side of a coding region in the four *Ilex* chloroplast genomes with DnaSP 6.0 software. The $\pi$ values of the four *Ilex* chloroplast genomes ranged from 0.00000 to 0.01667, with an average $\pi$ value of 0.00314. Based on a cutoff value for $\pi > 0.005$, we identified 33 highly variable regions out of the 626 windows analyzed (Figure 4 and Table S1). Twenty-eight of these regions (*trnH$^{GUG}$-psbA*, *trnK$^{UUU}$-matK*, *matK-rps16*, *rps16-trnQ$^{UUG}$*, *trnG$^{UCC}$*, *trnG$^{GCC}$-trnR$^{UCU}$*, *atpA-atpF*, *atpH-atpI*, *rpoC2*, *rpoB*, *rpoB-trnC$^{GCA}$*, *petN-psbM*, *trnT$^{GGU}$-psbD*, *trnS$^{UGA}$-psbZ*, *trnG$^{GCC}$-trnfM$^{CAU}$*, *psaA-ycf3*, *ycf3-trnS$^{GGA}$*, *trnT$^{UGU}$*, *trnL$^{UAA}$*, *ndhC-trnV$^{UAC}$*, *atpB-rbcL*, *psbE-petL*, *clpP*, *psbB-psbN*, *psbT-psbH*, *petB-petD*, *rpl36-rps8*, and *rpl16-rpl22*) were located in LSC regions, whereas five (*ndhF-rpl32*, *trnL$^{UAG}$-ccsA*, *ndhE*, *ndhA*, and *ycf1*) were located in SSC regions (Table S1). The Pi values of the 33 hypervariable loci ranged from 0.008 to 0.016.

Because DNA molecular markers can be used in HRM analysis, and regions with high Pi values do not necessarily have high discrimination rates for the four *Ilex* species, we tested sequence identity rates using the BLAST search tool of the NCBI database to select species-specific regions for the four *Ilex* species from the 33 highly variable regions with high Pi values. Accordingly, we used the ~600 bp nucleotide sequences of *I. integra* derived from each of the 33 hypervariable regions as a reference sequence to search the database Nucleotide Collection for *Ilex* Species using MegaBLASt in NCBI. As shown in Table S1, when *I. integra* was used as the reference sequence, each locus had a query coverage of 100% and 100% sequence identity. Only one region, *trnS$^{UGA}$-psbZ*, showed a clearly different percentage among the four *Ilex* species, with sequence identities of 98.35%, 98.36%, and 98.52% for *I. rotunda*, *I. cornuta*, and *I. x wandoensis*, respectively. Among the 32 other regions, the sequence identities between *I. cornuta* and *I. x wandoensis* were 100% in 20 regions (*trnH$^{GUG}$-psbA*, *trnG$^{GCC}$-trnR$^{UCU}$*, *atpA-atpF*, *atpH-atpI*, *rpoC2*, *rpoB*, *rpoB-trnC$^{GCA}$*, *petN-psbM*, *trnT$^{GGU}$-psbD*, *trnG$^{GCC}$-trnfM$^{CAU}$*, *trnT$^{UGU}$*, *trnL$^{UAA}$*, *ndhC-trnV$^{UAC}$*, *psbB-psbN*, *psbT-psbH*, *rpl36-rps8*, *rpl16-rpl22*, *ndhF-rpl32*, *ndhE*, and *ndhA*), and the sequence identities between *I. integra* and *I. x wandoensis* were 100% in seven regions (*trnK$^{UUU}$-matK*, *matK-rps16*, *trnG$^{UCC}$*, *ycf3-trnS$^{GGA}$*, *atpB-rbcL*, *clpP*, and *trnL$^{UAG}$-ccsA*). The sequence identities for three *Ilex* species (*I. integra*, *I. cornuta*, and *I. x wandoensis*) were also 100% in four regions (*rps16-trnQ$^{UUG}$*, *psaA-ycf3*, *psbE-petL*, and *ycf1*), while the sequence identities were 100% between *I. integra* and *I. cornuta* in *petB-petD*.

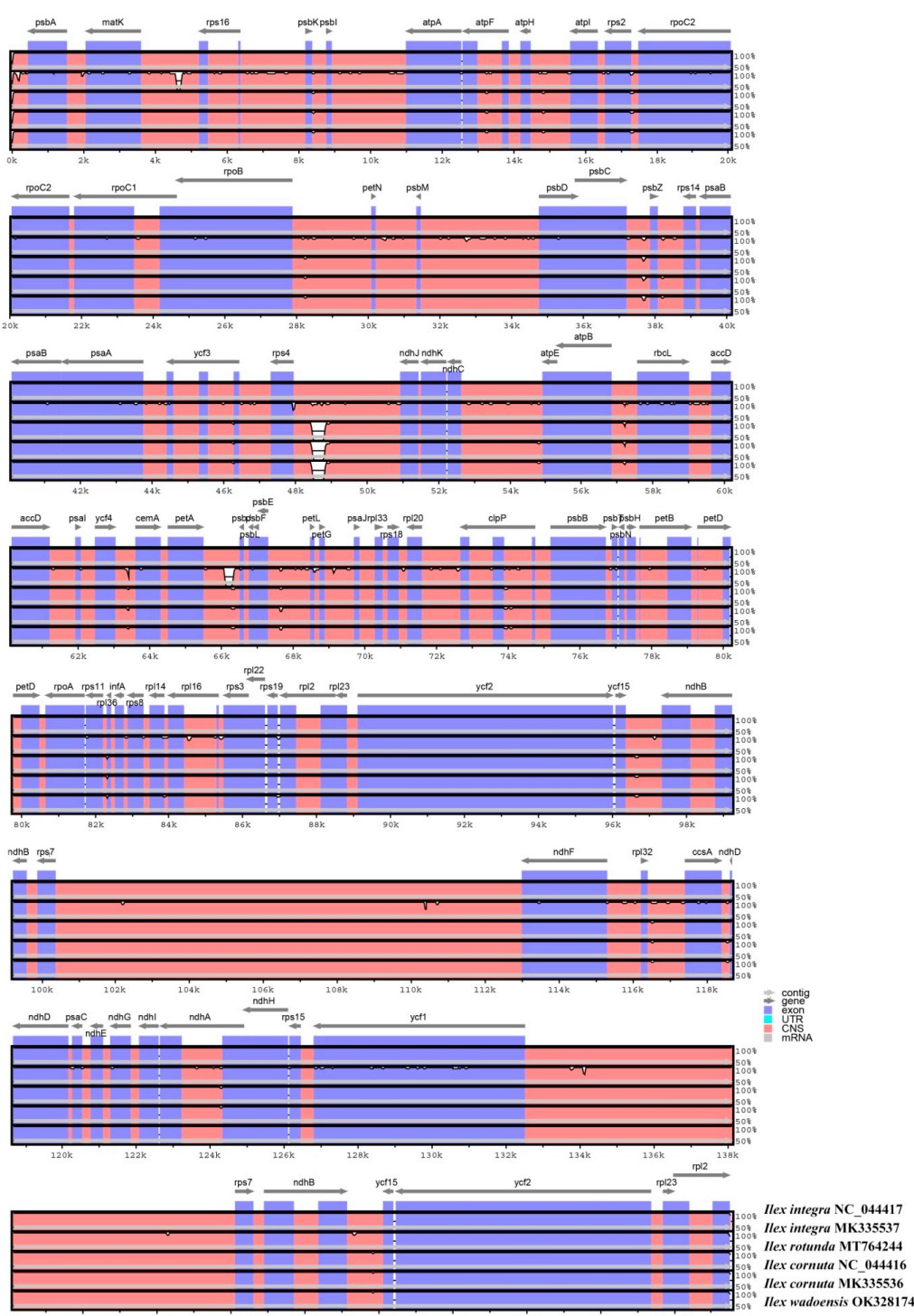

**Figure 3.** Comparison of the six chloroplast genome sequences using mVISTA. The cp genomes of the four *Ilex* species were compared with that of *I. integra* NC_044417. Blue blocks: Conserved genes, red blocks: conserved non-coding sequences. White represents regions with sequence variation among the four Ilex species.

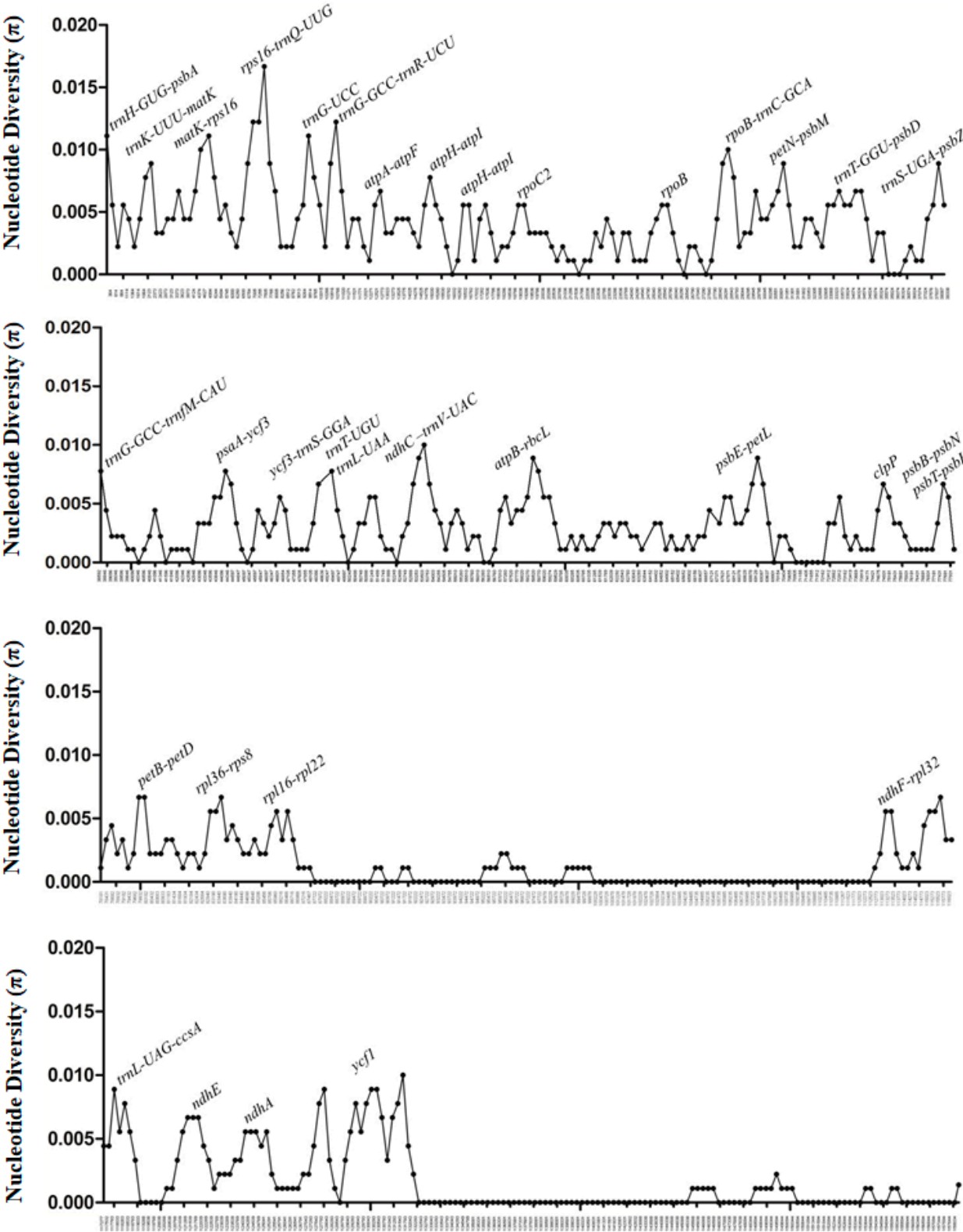

**Figure 4.** Sliding 600 bp window analysis of the six *Ilex* chloroplast genomes showing nucleotide diversity (π). Regions with π > 0.005 are indicated in the graph. *X*-axis, positions of the midpoints of windows; *y*-axis, nucleotide diversity of each window.

### 3.3. Microsatellite Genotyping of the Four Ilex Species Using HRM Analysis

In this study, we identified 33 variable regions by sliding window analysis of the complete chloroplast genomes of four *Ilex* species (Table S1). Regions with the highest levels of divergence ($\pi > 0.005$) commonly contained complex microsatellite loci, including SSRs, SNPs, and InDels. Among these, we chose a specific divergent region, the IGS region between $trnS^{UGA}$ and *psbZ*, which was a species-specific polymorphic site among *Ilex* species, to develop high-resolution molecular markers for the identification of these four species. The markers derived from $trnS^{UGA}$-*psbZ* loci were composed of mono-nucleotide repeat motifs such as A and C, as well as point mutations comprising T/A at position 83, A/T/G/C between positions 88 and 100, an inserted T and G at position 107, and A/C between positions 111 and 115 (Figure 4). We designed specific primers against the conserved regions and used them for HRM analysis (Table 3). The HRM curves for the target region $trnS^{UGA}$-*psbZ* reveal that *I. integra* could be visually separated from *I. rotunda*, *I. cornuta*, and *I. x wandoensis*, as these regions were highly characteristic for each species (yellow, red, green, and blue curves, respectively, in Figure 5). Sanger sequencing confirmed the identities of the 173-bp amplicon for I. integra, the 169-bp amplicon for I. rotunda, the 179-bp amplicon for I. cornuta, and the 178-bp amplicon for I. x wandoensis, whose size differences resulted from InDels of (A)n between nucleotides 88 and 104 and (C)n between nucleotides 111 and 122 (Figure 5). The Tm values of the four *Ilex* species and the melting profiles of the amplicons from the $trnS^{UGA}$-*psbZ* region are shown in Figure 6. In the HRM analysis, the $trnS^{UGA}$-*psbZ* loci produced unique melting curves, and the accessions were stratified into four species.

**Table 3.** Characteristics of the microsatellite marker used for HRM analysis.

| Locus | Region | Repeat Motif | Sequence (5′-3′) | Tm (°C) | Size (bp) |
|---|---|---|---|---|---|
| $trnS^{UGA}$-*psbZ* | IGS | (T/A)(A)$_4$(A/T/G/C)(A)$_n$(T)$_2$ (T/G)(C)$_2$(A)$_n$(C)$_n$ | TGCATGCCCATTTGTGAAA CATTTGATCCCTCTATCAGCCA | 60 | 170–180 |

**Figure 5.** Alignment of a subset of DNA fragments at the simple sequence repeat (SSR) locus $trnS^{UGA}$-*psbZ* in 15 *Ilex* chloroplast genomes. Numbers at the top indicate positions relative to the consensus sequence between 70 and 130 bp. Bold nucleotides in brackets indicate SSR motifs, and the number of repeats is shown. Dashes indicate gaps. Point mutations are highlighted with colored boxes; red: A, green: T, blue: G, and purple: C. Yellow boxes indicate two groups of insertions and deletions, which are considered to be a single mutational event.

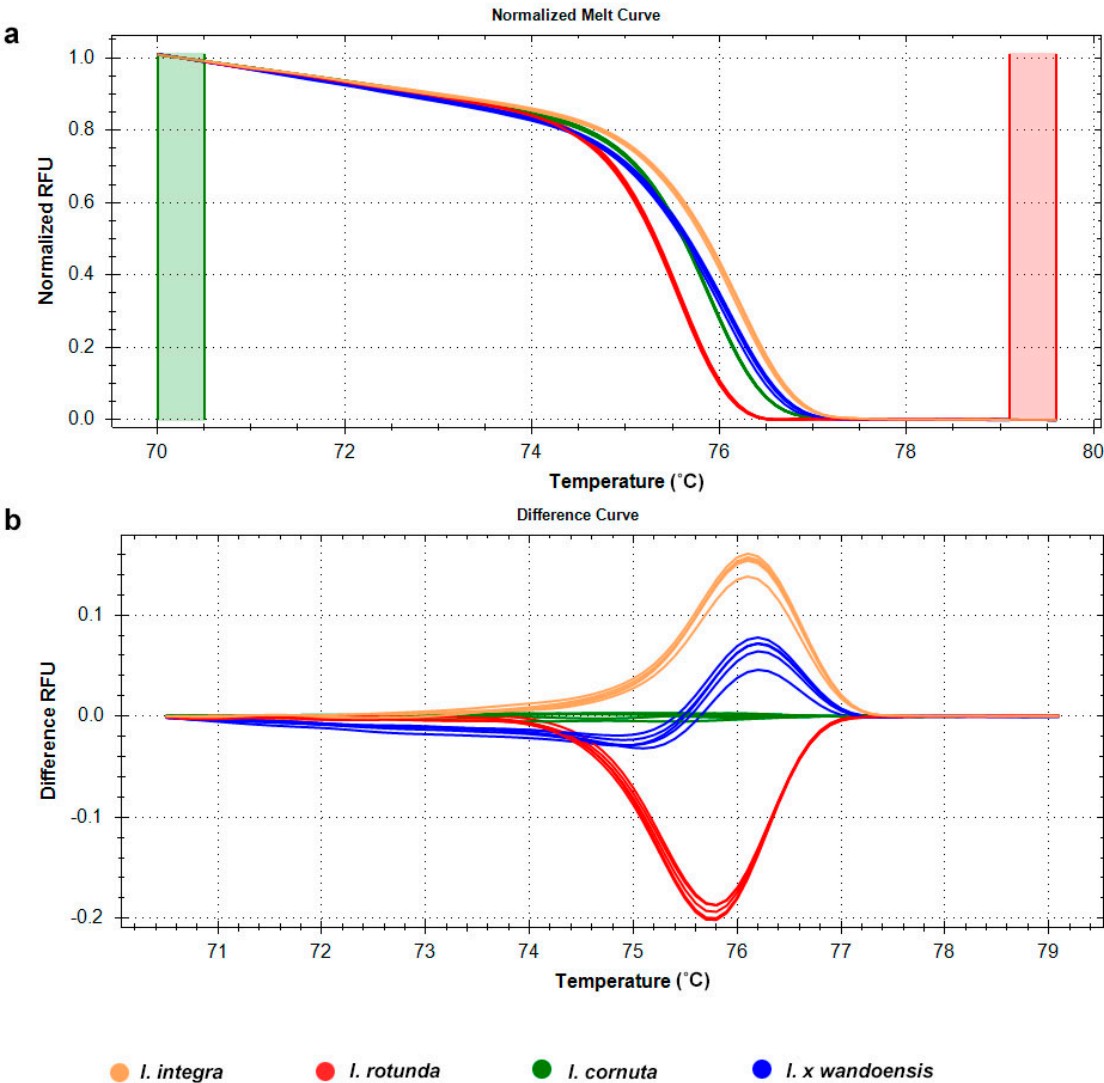

**Figure 6.** Microsatellite typing of the four *Ilex* species using HRM analysis with the SSR marker *trnS^{UGA}-psb*Z. (**a**) Normalized melting curves. (**b**) Difference plot curves.

## 4. Conclusions

The unequivocal identification and discrimination of similar species within the same genus in land plants, particularly species that serve as potential sources of secondary metabolites for the food and medicinal industries, has generally been a challenge for researchers and for collectors in the field. *I. integra*, which has similar biological activities as *I. paraguariensis* (Yerba mate), is a native species to Korea that has been attracting increasing attention for its potential use in the functional food and pharmaceutical industries [1,27,28]. Therefore, in the current study, we devised a method to rapidly and effectively distinguish *I. integra* from three native *Ilex* species, *I. rotunda*, *I. cornuta*, and *I. x wandoensis*, to facilitate the standardization and quality control of raw materials from this natural resource. DNA barcoding based on core DNA barcode regions can be used to confirm the identities of natural raw materials and to develop standard-quality herbal medicinal products for the healthcare marketplace. However, this method rarely provides accurate species identification since discriminating between closely related species within the same genus is difficult and often impossible. Moreover, the diagnostic features examined may not be sufficiently unique for a given species [17,29]. Super barcodes, which can distinguish closely related taxa through comparative analysis of the whole genome sequences of small organelles such as chloroplasts and mitochondria, have been proposed as an alternative to universal

DNA barcodes. However, it is difficult to establish super barcodes due to the costly, time-consuming research needed for their identification [17]. Such studies require customized protocols that use optimized tools to discriminate between species within each genus.

Many researchers have recently identified hypervariable regions between species by sliding window analysis of the entire chloroplast genomes of different species at the genus level [30–32]. These regions can be thought of as 'accurate DNA barcodes' or 'customized DNA barcodes' between species within each genus, a concept opposite from that of general universal plant DNA barcodes. Here, we retrieved the complete chloroplast genome sequences of four *Ilex* species from the NCBI database, aligned and compared them to identify hypervariable regions with species-specific genetic diversity. We identified 33 hypervariable regions in the four *Ilex* species, with an average nucleotide diversity value ($\pi$) of 0.011333 (>0.005) by sliding window analysis (600 sites). Although the average $\pi$ value for the 33 regions was higher, these regions did not show high species-specific divergence for the four individual *Ilex* species. Therefore, as a more efficient approach to species-specific marker selection, we visually selected specific sites for each of the four *Ilex* species from the 33 hypervariable regions using the NCBI BLAST search algorithm. Only one region, the $trnS^{UGA}$-*psbZ* IGS region, contained specific SSR markers that discriminated between each of the four *Ilex* species. We confirmed that this region can specifically distinguish not only the four *Ilex* species but also other species within the same genus.

HRM analysis using the microsatellite marker derived from the $trnS^{UGA}$-*psbZ* IGS SSR region successfully distinguished the four different *Ilex* species. This method is simple enough for use in identifying raw materials of *I. integra* by the medicinal herb industry. This technique was sensitive enough to identify amplicons based on changes in 2 mono-nucleotide and point mutations ($trnS^{UGA}$-*psbZ* IGS), highlighting its potential for performing accurate large-scale screening. To enhance the development and commercialization of functional plant materials, there is a demand for an accurate method that can rapidly and easily discriminate among different species within the same genus. Such a method must be certified and standardized for the authentication of raw plant materials from cultivated or wild-harvested materials at the source or prior to manufacturing. Although universal core DNA barcodes are limited in their ability to discriminate among species, coupling DNA barcodes with HRM analysis has emerged as an excellent alternative technological platform for the precise identification of medicinal plant materials. Therefore, we suggest that the improved species-specific method designed in this study involving the combination of HRM analysis and the use of a chloroplast genome-based "customized DNA barcode" could be optimized for different species within the same genus.

**Supplementary Materials:** The following supporting information can be downloaded at: https://www.mdpi.com/article/10.3390/f13101718/s1, Table S1: 33 regions of the four *Ilex* cp genome sequence alignment (window of 600 sites) with significantly high ($\pi > 0.006$) nucleotide diversity and different sequence identities (%) based on the nucleotide sequence of *I. integra*.

**Author Contributions:** Study conception and design, D.B. and Y.K.; material preparation and experiments, D.-R.O. and Y.-J.K.; data analysis, Y.K. and K.-N.O.; writing—original draft preparation, review, and editing, Y.K.; project administration and funding acquisition, D.B. All authors have read and agreed to the published version of the manuscript.

**Funding:** This research was funded by the Ministry of Trade, Industry and Energy (MOTIE, Korea) [No. P0017660].

**Institutional Review Board Statement:** This study was approved by the institutional review board of the author's affiliated research institute (protocol code 001-01 March 2021).

**Informed Consent Statement:** Not applicable.

**Data Availability Statement:** The geographical locations of the collected *Ilex* samples in South Korea and the *Ilex* genome sequences with GenBank accession numbers are included in the manuscript.

**Acknowledgments:** This paper is dedicated to the memories of Heungsu Kim and Jaeman Kim for inspiring us, mentoring our research, and teaching us to be scientifically curious, inquisitive inventors as well as scientists with a passion for plant biology.

**Conflicts of Interest:** The authors declare no conflict of interest.

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
