# Peer review of "Chloroplast Microsatellite-Based High-Resolution Melting Analysis for Authentication and Discrimination of Ilex Species"

_forests, doi:10.3390/f13101718_

Round 1
Reviewer 1 Report
I suggest not abbreviating chloroplast as cp. It is a single word and better to spell it completely. " The lengths of the large single-copy (LSC) region ranged from 86,607 bp (I. x 166 wandoensis) to (I. rotunda)," here value for the range is missing. " The lengths of these cp genomes ranged from 157,216 bp (I. x wandoensis) to 157,780 bp (I. rotunda). The lengths of the large single-copy (LSC) region ranged from 86,607 bp (I. x wandoensis) to (I. rotunda), while the small single-copy (SSC) regions ranged from 18,426 bp (I. integra) to 18,436 bp (I. rotunda). The pairs of inverted repeat (IR) regions ranged from 52,182 bp (I. cornuta and I. x wandoensis) to 52,250 bp (I. rotunda)." - this is too much descriptive and redundant with the table. Is there any information on the mitochondrial genomes of these species?Author Response
Response to Reviewers
We appreciate the reviewers for your precious time in reviewing our paper and providing valuable comments. It was your valuable and insightful comments that led to possible improvements in the current version. The authors have carefully considered the comments and tried our best to address every one of them. We hope the manuscript after careful revisions meet your high standards. The authors welcome further constructive comments if any. Below we provide the point-by-point responses.
Reviewer #1
[General Comment] I suggest not abbreviating chloroplast as cp. It is a single word and better to spell it completely.
Response: We sincerely appreciate the reviewer's comments. We have made revisions accordingly.
[Comment 1] The lengths of the large single-copy (LSC) region ranged from 86,607 bp (I. x wandoensis) to (I. rotunda)," here value for the range is missing.
Response: Thanks for your kind reminders. We have made revisions accordingly.
The lengths of the large single-copy (LSC) region ranged from 86,607 bp (I. x wandoensis) to 87,094 bp (I. rotunda), while the small single-copy (SSC) regions ranged from 18,426 bp (I. integra) to 18,436 bp (I. rotunda).
[Comment 2] " The lengths of these cp genomes ranged from 157,216 bp (I. x wandoensis) to 157,780 bp (I. rotunda). The lengths of the large single-copy (LSC) region ranged from 86,607 bp (I. x wandoensis) to (I. rotunda), while the small single-copy (SSC) regions ranged from 18,426 bp (I. integra) to 18,436 bp (I. rotunda). The pairs of inverted repeat (IR) regions ranged from 52,182 bp (I. cornuta and I. x wandoensis) to 52,250 bp (I. rotunda)." - this is too much descriptive and redundant with the table.
Response: Thanks for your kind reminders. We removed the table 2 according to the reviewer’s comment.
[Comment 3] Is there any information on the mitochondrial genomes of these species?
Response: Searching for genome database including NCBI GenBank, there was no information on the mitochondrial genomes of Ilex species except for Ilex pubescens.
Reviewer 2 Report
The study is interesting, well-prepared and adequately described. The highly advanced molecular methods of DNA analyses were used and resulted in clear, transparent output (Fig. 6B). I have a few questions, as follows:
Line 29
Ilex species occur also in Europe, Africa and Australia. For example, Ilex aquifolium occurs in Europe and northern Africa, and was introduced to North America and then, all over the world. Would be interesting to add non-native to Korea Ilex species such as I. aquifolium to the study.
Then you can compare Ilex species of different origins. Origins or rather the clads inside the genus Ilex probably influence the results you obtained.
Picked Ilex species
Four Ilex taxa were analysed in the presented study: three ilex species (I. integra, I. rotunda, I. cornuta) and one hybrid (Ilex x wandoensis, hybrid of I. cornuta and I. intergra) out of >660 Ilex species available.
Analyzed Ilex species belong to 3 distinct clads named C3 (I. integra), C4 (I. cornuta) and E7 (I. rotunda). Among them, clads C3 and C4 are relatively close related (what's proved by hybridization among these species resulted in Ilex x wandoensis), but clad E7 is distanced in relation to clads C3 and C4.
[Please, look at Yao et al. 2022: https://doi.org/10.3390/f13010094 ]
Therefore my conclusion is: could the Authors assign the obtained results to the species-specific features of distinct Ilex species, or to the clad-specific features?
The results presented by the Authors contain only one representative per single clade, with two representatives for one group of clads named C (I. integra, I. cornuta and their hybrid) and only one representative of second group of clads named E (I. rotunda), with no representation for other clads [according to the phylogeny provided by Yao et al. 2022]. Moreover, clads C, D and E are represented by 6,5 and 9 subclades respectively. Finally, the Authors analyzed 3 species from 3 subclades out of more than 660 Ilex species classified into more than 20 subclades.
I suppose that's not enough to develop any species-specific DNA markers, but I would like to read other reviews and the other reviewers' opinions.
Author Response
Response to Reviewers
We appreciate the reviewers for your precious time in reviewing our paper and providing valuable comments. It was your valuable and insightful comments that led to possible improvements in the current version. The authors have carefully considered the comments and tried our best to address every one of them. We hope the manuscript after careful revisions meet your high standards. The authors welcome further constructive comments if any. Below we provide the point-by-point responses.
Reviewer #2
[General Comment] The study is interesting, well-prepared and adequately described. The highly advanced molecular methods of DNA analyses were used and resulted in clear, transparent output (Fig. 6B). I have a few questions, as follows:
Response: We sincerely appreciate the reviewer's comments.
[Comment 1] Ilex species occur also in Europe, Africa and Australia. For example, Ilex aquifolium occurs in Europe and northern Africa, and was introduced to North America and then, all over the world. Would be interesting to add non-native to Korea Ilex species such as I. aquifolium to the study.
Then you can compare Ilex species of different origins. Origins or rather the clades inside the genus Ilex probably influence the results you obtained.
Response: Thanks for your kind reminders. If Ilex aquifolium species is also included in our study, this manuscript might be more meaningful. Unfortunately, the chloroplast genome information for this species could not be found in the NCBI GenBank database. In addition, it is difficult to collect the tree or samples because this species is less well known than Ilex cornuta in Korea. Due to the limited Ilex species collection, our results were derived from native-species in Korea. If we collected complete chloroplast genome for Ilex species distributed worldwide, further research may derive phylogenetic analysis with origin and clades for Ilex genus.
[Comment 2] Four Ilex taxa were analysed in the presented study: three ilex species (I. integra, I. rotunda, I. cornuta) and one hybrid (Ilex x wandoensis, hybrid of I. cornuta and I. intergra) out of >660 Ilex species available.
Analyzed Ilex species belong to 3 distinct clads named C3 (I. integra), C4 (I. cornuta) and E7 (I. rotunda). Among them, clades C3 and C4 are relatively close related (what's proved by hybridization among these species resulted in Ilex x wandoensis), but clad E7 is distanced in relation to clads C3 and C4.
[Please, look at Yao et al. 2022: https://doi.org/10.3390/f13010094 ]
Therefore my conclusion is: could the Authors assign the obtained results to the species-specific features of distinct Ilex species, or to the clad-specific features?
Response: Thanks for your kind reminders. This result clearly has limitation for broad phylogenetic analysis of Ilex genus. However, as shown HRM results (Figure 6), it is meaningful as a molecular marker that can distinguish three species (I. integra, I. cornuta, and I. rotunda) and hybrids (Ilex x wandoensis) by limiting the Ilex species distributed in Korea.
[Comment 3] The results presented by the Authors contain only one representative per single clade, with two representatives for one group of clads named C (I. integra, I. cornuta and their hybrid) and only one representative of second group of clads named E (I. rotunda), with no representation for other clads [according to the phylogeny provided by Yao et al. 2022]. Moreover, clads C, D and E are represented by 6,5 and 9 subclades respectively. Finally, the Authors analyzed 3 species from 3 subclades out of more than 660 Ilex species classified into more than 20 subclades.
I suppose that's not enough to develop any species-specific DNA markers, but I would like to read other reviews and the other reviewers' opinions.
Response: Thanks for your kind reminders. We agree with the reviewers’ comments. Although this result is limited, as shown results of the HRM analysis (Figure 6), we founded the trnSUGA-psbZ loci produced unique melting curves, and the accessions were stratified into four species. As shown Figure 5, we confirmed that there was a significant discrimination in the microsatellite marker region between 15 Ilex species.